# Protein Extract of a Probiotic Strain of *Hafnia alvei* and Bacterial ClpB Protein Improve Glucose Tolerance in Mice

**DOI:** 10.3390/ijms241310590

**Published:** 2023-06-24

**Authors:** Vasiliy A. Zolotarev, Vladimir O. Murovets, Anastasiya L. Sepp, Egor A. Sozontov, Ekaterina A. Lukina, Raisa P. Khropycheva, Nina S. Pestereva, Irina S. Ivleva, Mouna El Mehdi, Emilie Lahaye, Nicolas Chartrel, Sergueï O. Fetissov

**Affiliations:** 1Pavlov Institute of Physiology, Russian Academy of Sciences, 199034 Saint-Petersburg, Russia; zolotarevva@infran.ru (V.A.Z.); murovetsvo@infran.ru (V.O.M.); seppal@infran.ru (A.L.S.); sozontovea@infran.ru (E.A.S.); lukinaea@infran.ru (E.A.L.); hropy4ewa@yandex.ru (R.P.K.); 2Institute of Experimental Medicine, 197376 Saint-Petersburg, Russia; pesterevans@yandex.ru (N.S.P.); i.s.oblamskaya@mail.ru (I.S.I.); 3Inserm UMR1239 Laboratory, Team: Regulatory Peptides—Energy Metabolism and Motivated Behavior, University of Rouen Normandie, 76130 Mont-Saint-Aignan, France; elmehdi.mouna@gmail.com (M.E.M.); emilie.lahaye1@univ-rouen.fr (E.L.); nicolas.chartrel@univ-rouen.fr (N.C.)

**Keywords:** glucose metabolism, glucose tolerance, insulin, sweet taste, probiotics, *Hafnia alvei*, ClpB, neuropeptides

## Abstract

A commercial strain of *Hafnia alvei* (*H. alvei*) 4597 bacteria was shown to reduce food intake and promote weight loss, effects possibly induced by the bacterial protein ClpB, an antigen-mimetic of the anorexigenic α-melanocyte-stimulating hormone. A decrease in the basal plasma glucose levels was also observed in overweight fasted humans and mice receiving *H. alvei*. However, it is not known whether *H. alvei* influences sweet taste preference and whether its protein extract or ClpB are sufficient to increase glucose tolerance; these are the objectives tested in the present study. C57BL/6J male mice were kept under standard diet and were gavaged daily for 17 days with a suspension of *H. alvei* (4.5 × 10^7^ CFU/animal) or with *H. alvei* total protein extract (5 μg/animal) or saline as a control. Sweet taste preference was analyzed via a brief-access licking test with sucrose solution. Glucose tolerance tests (GTT) were performed after the intraperitoneal (IP) or intragastric (IG) glucose administration at the 9th and 15th days of gavage, respectively. The expression of regulatory peptides’ mRNA levels was assayed in the hypothalamus. In another experiment performed in non-treated C57BL/6J male mice, effects of acute IP administration of recombinant ClpB protein on glucose tolerance were studied by both IP- and IG-GTT. Mice treated with the *H. alvei* protein extract showed an improved glucose tolerance in IP-GTT but not in IG-GTT. Both groups treated with *H. alvei* bacteria or protein extract showed a reduction of pancreatic tissue weight but without significant changes to basal plasma insulin. No significant effects of *H. alvei* bacteria or its total protein extract administration were observed on the sweet taste preference, insulin tolerance and expression of regulatory peptides’ mRNA in the hypothalamus. Acute administration of ClpB in non-treated mice increased glucose tolerance during the IP-GTT but not the IG-GTT, and reduced basal plasma glucose levels. We conclude that both the *H. alvei* protein extract introduced orally and the ClpB protein administered via IP improve glucose tolerance probably by acting at the glucose postabsorptive level. Moreover, *H. alvei* probiotic does not seem to influence the sweet taste preference. These results justify future testing of both the *H. alvei* protein extract and ClpB protein in animal models of diabetes.

## 1. Introduction

Recent studies have shown the association between gut microbiota and regulation of glucose metabolism in both physiological conditions and during metabolic diseases [1]. Targeting gut microbiota, for example, with probiotics, appears to be a promising adjunctive therapy for altered glucose metabolism, but the most efficient species, the mode of delivery, and the underlying molecular mechanisms remain to be better determined [2]. Indeed, several molecular pathways may link gut microbiota with the host mechanisms of glucose and energy metabolism regulation. Gut bacteria may enable the host to extract energy from indigestible food and catalyze the production of short-chain fatty acids (SCFA) that elicit local and systemic regulatory responses, including stimulation of the enteroendocrine cells [3]. In turn, the secretion of gut hormones such as peptide YY (PYY), glucagon-like peptide-1 (GLP-1), etc., will activate endocrine and brain targets regulating energy balance and glucose homeostasis [4].

Gut-bacteria-derived protein mimetics of regulatory peptides may also link gut microbiota to the host neuroendocrine regulation of energy balance. For instance, caseinolytic protease B (ClpB) is a 96 kDa protein produced by the Enterobacteriaceae family, which was identified as the conformational antigen mimetic of α-melanocyte-stimulating hormone (α-MSH) [5]. α-MSH is an anorexigenic peptide of the melanocortin (MC) system, critically involved in the regulation of appetite, body weight, and glucose homeostasis [6]. Although the brain appears to be the primary site of α-MSH’ regulatory effects on energy metabolism, α-MSH has many peripheral targets, including the gut, where it may stimulate PYY secretion by enteroendocrine cells expressing MC type 4 receptor (MC4R) [7]. ClpB is also able to stimulate PYY release from rat intestinal mucosa [8,9].

Provision of obese mice with ClpB-producing strains of bacteria, such as *Escherichia coli* (*E. coli*) and *Hafnia alvei* (*H. alvei*), reduces food intake and body weight gain [10,11]. Importantly, deletion of the ClpB gene in *E. coli* significantly diminished the anorexigenic and body-weight-lowering effects [10]. *H. alvei* has long been used for the fermenting of soft cheeses and is a relatively rare commensal species present in the gut microbiota of healthy subjects, an abundance of which correlates negatively with body mass index [10]. A clinical study of *H. alvei* supplementation in overweight humans undergoing a mild hypocaloric diet, led to a greater loss of body weight and body fat content as well as an increase in fulness perception as compared to a placebo [12]. Another finding of this study was that taking *H. alvei* by overweight but otherwise healthy, non-diabetic subjects resulted in a slight, but significant decrease in basal glycemia. A decrease in basal glycemia and an improved glucose tolerance was also observed in *H. alvei*-treated overweight mice with high-fat-diet-induced obesity [11].

These data suggest that *H. alvei*, as a live probiotic, can be used as a food supplement for the management of body weight and for improving glucose tolerance. It is, however, still unknown whether the protein extract of *H. alvei* or protein ClpB is sufficient to improve glucose tolerance in non-obese and non-diabetic animals. Moreover, the natural orosensory as well as visceral preference for sugar intake may constitute a risk factor for metabolic diseases [13,14]. Indeed, in obese subjects, sensitivity to sweet substances is reduced, leading to overeating of sugar-rich food [15,16]. It is, hence, of relevance, in order to decipher potential mechanisms of action of *H. alvei* on glucose metabolism, to investigate whether these bacteria may influence oral sweet-taste sensitivity and sucrose preference.

Thus, in the present study, we analyzed the effects of chronic provision via intragastric gavage of *H. alvei* bacteria or their total protein extract on sucrose-intake preference and glucose tolerance in healthy, lean mice. For the possible underlying mechanisms, the level of expression of several regulatory peptides’ mRNA was assayed in the hypothalamus. The choice of the gene targets was based on the known role of each selected hypothalamic peptide in the regulation of both energy and glucose metabolism [17,18,19,20,21]. Additionally, glucose tolerance in mice was studied after acute intraperitoneal (IP) administration of a recombinant ClpB protein.

## 2. Results

### 2.1. Experiment 1

#### 2.1.1. Sweet Preference Test

To study sweet preference, the brief-access licking test (BALT) was performed on days 11 and 12 of the supplementation with *H. alvei* bacteria or bacterial protein extract (Figure 1).

All mice showed a dose-dependent increase in sucrose solution licking rate (number of licks per 5 sec of access period). Pretreatment with a suspension of *H. alvei* or its protein extract did not change the licking rate responses to either the low or high range of sucrose concentrations (Figure 2).

#### 2.1.2. Baseline Glucose and Body Composition

In non-fasted mice, the basal plasma glucose concentration measured on the 17th day of probiotic administration was lower after pretreatment with *H. alvei* bacteria than in animals receiving *H. alvei* protein extract. An intermediate basal glucose level was observed in the control group (one-way ANOVA: F(2, 32) = 4.23, *p* < 0.024; *H. alvei* bacteria vs. *H. alvei* protein extract *p* < 0.0181, Tukey HSD post hoc test, Figure 3a).

Mice displayed a stable body weight during the experiment without significant effects of the probiotic treatment on Day 17, (24.66 ± 0.33 and 24.23 ± 0.43 g after protein extract or suspension of *H. alvei*, respectively, vs. 24.33 ± 0.33 g for Control, one-way ANOVA: F(2, 32) = 0.58, *p* > 0.69). Body composition analysis revealed no significant effect of the treatment on body fat content (Figure 3b), relative liver mass (Figure 3d), and glycogen content in the liver and muscle (Figure 3e,f). However, the relative mass of the pancreas was reduced by about 21% in mice receiving *H. alvei* protein extract and by about 13% in mice receiving *H. alvei* bacteria as compared to the control group (one-way ANOVA: F(2, 32) = 5.25, *p* < 0.011; Control vs. *H. alvei* extract *p* < 0.0086, control vs. *H. alvei* bacteria *p* < 0.05, Tukey HSD post hoc tests, Figure 3c).

#### 2.1.3. Glucose Tolerance Tests

Treatment of mice with probiotics potentiated plasma glucose clearance when glucose was injected intraperitoneally (IP-GTT) as confirmed via two-way ANOVA: effect of probiotic F(2, 31) = 4.50, *p* < 0.02; effect of time F(7, 217) = 118.94, *p* < 0.00001; probiotic × time F(14, 217) = 3.22, *p* < 0.0002. Post-hoc comparisons showed that only the bacterial protein extract improved glucose tolerance (Figure 4a). The effect of gavage with bacterial protein was further analyzed via comparisons of AUCs, showing a significant glucose-lowering effect of the *H. alvei* protein extract (Figure 4a). In IG-GTT, pretreatment with either the bacterial suspension or the protein extract of *H. alvei* did not show significant effect on plasma glucose levels (Figure 4b).

Whether probiotics may influence the so-called ‘incretin effect’, i.e., involve secretion of insulin-stimulating intestinal hormones resulting in enhanced anti-hyperglycemic response, was estimated by comparing the plasma glucose levels obtained during IG- and IP-GTT [22]. Figure 5 illustrates such an ‘incretin effect’ presented as % of differences between IP and IG vs. 100% IG glucose levels as a maximal “incretin effect”. Mice receiving *H. alvei* bacterial suspension displayed a visibly elevated “incretin effect” (ANOVA: effect of probiotic F(2, 32) = 4.48, *p* < 0.02; effect of time F(6, 192) = 3.22, *p* < 0.005; probiotic × time F(12, 192) = 1.02, *p* > 0.42), which was significant for AUC as compared to the *H. alvei* protein group (Figure 5).

#### 2.1.4. Plasma Insulin and Insulin Tolerance Test

Pretreatment with a *H. alvei* bacterial suspension or a bacterial protein extract did not significantly alter basal plasma insulin level (one-way ANOVA: F(2, 26) = 1.02, *p* > 0.37; *p* > 0.34~0.90, Tukey HSD post hoc test—Figure 6a, *p* > 0.36), although a tendency of a lower level was observed in mice treated with *H. alvei* bacteria (Student’s t-test *p* = 0.08, vs. Control). During the ITT in non-fasted mice, injection of insulin (1 U/kg, IP) induced a rapid (within 15 min) decrease in plasma glucose concentration with further elevation during the following 2 h. Probiotic administration had no significant effect (ANOVA: effect of probiotic F(2, 27) = 0.32, *p* > 0.72; effect of time F(3, 81) = 231.42, *p* < 0.0001; probiotic × time F(3, 81) = 1.32, *p* > 0.25) on the insulin-induced hypoglycemic response (Figure 6b).

#### 2.1.5. Regulatory Peptide mRNA Levels in the Hypothalamus

Treatment with *H. alvei* probiotic preparations did not significantly modify the expression levels of mRNA in the hypothalamus for any of the regulatory peptide studied (Figure 7).

### 2.2. Experiment #2 (Exp#2)

#### ClpB Administration

To analyze whether ClpB may have a direct effect on plasma glucose and glucose tolerance, recombinant ClpB protein (10 μg) was given via a single IP injection to C57Bl/6 male mice maintained on a standard diet. ClpB administration in glucose-non-stimulated mice resulted in a slight decrease in basal levels of plasma glucose which was significant at the 105 min time-point after ClpB injection (Figure 8a). During the IP-GTT, ClpB administration 30 min before IP glucose injection, resulted in a significantly lower peak of plasma glucose as compared to the PBS-injected control group (Figure 8b). No significant effects of ClpB administration on plasma glucose during the IG-GTT one week later in the same mice were observed (Figure 8c). Estimation of the ‘incretin effect’ was performed using the same approach described above for *H. alvei* probiotic treatment, *i.e*., by calculating the percentage of difference between IP and IG levels of ‘incretin effect’ in its ability to increase glucose tolerance. It revealed that in its ability to improve glucose tolerance, IP ClpB does not involve the ‘incretin effect’ (Figure 8d).

## 3. Discussion

The main finding of the present study is the first demonstration of the abilities of the protein extract of *H. alvei* and the enterobacterial ClpB protein to increase glucose tolerance in lean, non-diabetic mice. On the contrary, we did not find any effect of *H. alvei* supplementation on the sweet taste preference in mice.

The rationale of testing *H. alvei* protein extract is to better understand the mechanisms underlying the slightly lower blood glucose levels found in healthy, overweight subjects taking *H. alvei* bacteria as a probiotic food supplement vs. the placebo group [12]. Moreover, protein preparations of *H. alvei* may potentially constitute a complementary approach for the prevention of diabetes and obesity through the preferential targeting of a mechanism involved in the regulation of glucose metabolism, which is different from the bacterial suspension. Indeed, while proteins are digested and absorbed in the small intestine, live bacteria may reach the large intestine where they can transiently persist and interact with the intestinal epithelium and enteroendocrine cells. The latter, are typically involved in the so-called incretin effect triggering secretion of insulin. This effect largely depends on the two insulinotropic hormones glucose-dependent insulinotropic peptide (GIP) and GLP-1 released in response to food ingestion from enteroendocrine K-cells or L-cells, respectively which have a direct stimulatory effect on pancreatic β-cells [23]. According to the ‘incretin’ concept, oral ingestion of glucose is more potent in stimulating insulin secretion than intravenous administration [24]. The combined action of incretins is believed to account for about 50% of the total insulin secretory response after a meal [25]. 

In this study, we found that mice receiving *H. alvei* protein extract showed an increased glucose tolerance in the IP-GTT but not in IG-GTT performed one week later. These results demonstrate the ability of *H. alvei* protein to promote an anti-hyperglycemic response probably involving the postabsorptive glucose targets. Furthermore, an estimation of the incretin effect in these mice also revealed that it did not contribute to the anti-hyperglycemic action of the *H. alvei* protein extract. These results suggest that some bacterial proteins derived from the total protein extract of *H. alvei* may stimulate insulin secretion independently from the intestinal incretins. Because such an effect was not present in mice receiving the *H. alvei* bacteria, it suggests that live bacteria do not release a sufficient amount of proteins in the upper gut, or that such bacterial proteins were not intended for secretion. Nevertheless, in the lower gut, after their multiplication, bacteria may release enough protein via either secretion or during natural lysis for the activation of enteroendocrine cells, triggering the incretin effect. Indeed, an enhanced incretin effect was observed in mice receiving *H. alvei* bacteria and these mice also displayed slightly reduced basal plasma glucose concentration. These results obtained in lean, non-diabetic mice are consistent with the previously reported mild anti-hyperglycemic effect of *H. alvei* bacteria supplementation in overweight humans and mice [11,12]. In the present experiment of supplementation of *H. alvei* in lean mice, no changes in body weight and body fat content were observed, suggesting that the lean phenotype is less sensitive to the satietogenic and body weight lowering effects of these bacteria previously shown in obese mice [10]. No significant changes of the mRNA expression levels of several neuropeptides, further support the view that the treatment did not influence the hypothalamic homeostatic mechanism of the energy metabolism regulation. It also suggests that the anti-hyperglycemic effect of the *H. alvei* protein treatment preferentially involves glucose-sensitive extrahypothalamic sites.

It is possible that the anti-hyperglycemic effect of *H. alvei* proteins may involve melanocortin (MC)-peptide-sensitive peripheral targets. This possibility is based on the production by *H. alvei* of ClpB protein, which was shown to display molecular mimicry with α-MSH, an anorexigenic peptide [5]. Importantly, while such mimicry is present in the ClpB protein produced by both *H. alvei* and in *E. coli* bacteria, ClpB expression is necessary to induce an anorexigenic effect by *E. coli* [10]. Moreover, ClpB IP injection decreases food intake in lean and obese mice [9]. In the present study, we found that a pharmacological dose of the recombinant ClpB, delivered to lean mice via IP injection, caused a decrease in basal plasma glucose and also lowered the glucose peak during the IP-GTT. Interestingly, injection of ClpB did not modify the glucose response in the IG-GTT, making the results of the peripheral ClpB injections similar to the effects of *H. alvei* protein supplementation. Moreover, ClpB injection did not visibly induce any incretin effect. These data suggest that ClpB protein may be at least one active substance from the total protein extract of *H. alvei*, involved in its anti-hyperglycemic effect. 

Potential ClpB targets, relevant to its anti-hyperglycemic effect, may include MC4R expressing cholinergic parasympathetic neurons, in which activation by MC ligands attenuates hyperglycemia [26]. Parasympathetic neurons innervate the pancreas and targeted activation of parasympathetic cholinergic intrapancreatic ganglia stimulates insulin secretion and increases glucose tolerance [27]. MC4R expressed by islet β-cells may also directly mediate a stimulatory effect of MC peptides on insulin secretion [28]. In contrast, hepatic glucose production is not likely to be a ClpB target since the MC4R activation of the liver innervation increases plasma glucose levels [29]. We also did not find any effect of treatment on liver mass and liver and muscle glycogen content. Thus, while the possibility of a direct binding and activation of MC4Rs by ClpB remains to be determined, we may speculate that the MC-receptor-expressing parasympathetic neurons innervating the pancreas as well as pancreatic β-cells could be the primary targets for ClpB and *H. alvei* proteins, underlying their anti-hyperglycemic effects after glucose absorption. Decreased tissue mass of the pancreas in the group receiving *H. alvei* protein extract is in favor of this conclusion, suggesting an improved regulation of insulin secretion by the ‘quality’ but not the ‘quantity’ of endocrine tissue.

Our results, showing the efficacy of *H. alvei* protein extract in improving glucose tolerance, are not the first demonstration of an effect of a bacterial protein on glucose homeostasis. For instance, an oral daily gavage (3 μg/day for 5 weeks) with Amuc_1100, a 32-kDa protein of *Akkermansia muciniphila*, in high-fat-diet-fed obese mice increased their glucose tolerance in the oral GTT [30]. These results rather suggest an incretin effect of this bacterial protein. Moreover, another 84-kDa protein named P-9, produced by *A. muciniphila*, was shown to activate GLP-1 secretion [31]. Thus, it possible that different bacterial proteins produced by several commensal species of gut microbiota participate in the host regulation of glucose metabolism, involving different molecular pathways. Furthermore, such protein-mediated specific pathways may be complementary and contribute to a more general link between gut bacteria and glucose metabolism, for example, involving SCFA and some other common bacterial metabolites [31].

The study has shown that *H. alvei* bacteria and their protein extract did not influence the sweet taste preference in mice. Sweet taste detection is mediated by the heterodimer T1R2/T1R3 taste receptors and glucose transporters at both the oral and intestinal levels [32,33]. However, the cephalic reflex on glucose ingestion resulting in insulin secretion was found to be induced only from the oral cavity [34]. Nevertheless, the rewarding value of glucose, i.e., influencing its intake is also triggered from the portal vein, i.e., involving the postabsorptive glucose sensors [35]. Activation of sweet taste receptors in the gut induces the incretin effect including GLP-1 secretion [36]. In the brain, the MC system is involved in the regulation of the sweet taste preference; for instance, application of a MC3/4R agonist has been shown to inhibit the licking response to sucrose [37]. Microorganisms may also modulate intestinal chemosensation, resulting in changes in the gut’s ability to detect and absorb sugars [38]. Nevertheless, the present study revealed that increased glucose tolerance induced by *H. alvei* protein extract supplementation does not involve a decrease in sweet taste preference, further favoring the role of the parasympathetic system as discussed above.

## 4. Materials and Methods

### 4.1. Bacterial Culture

*Hafnia alvei* 4597 was obtained from EnteroSatys^®^ caps (Targedys, Longjumeau, France) and isolated on Luria–Bertani (LB) agar. A single colony was incubated in 10 mL of LB broth overnight at 37 °C with orbital shaking (140 rpm). A fraction of resulting culture (0.2 mL) was diluted at 10^7^ CFU/mL in 200 mL (Erlenmeyer flask) of LB broth and incubated at 37 °C with orbital shaking (140 rpm) for 24h. After a 6 h incubation, which corresponded to the beginning of the stationary growth phase verified by the monitoring of optical density, the culture was stopped for sample collection. The bacterial number was determined via seeding on LB agar; 5 mL of sample contained 3.5 × 10^8^ CFU. Samples were diluted in 0.9% NaCl to 4.5 × 10^7^ in 200 µL, a single dose per mouse, aliquoted and stored at −20 °C for further use.

### 4.2. Bacterial Protein Extraction

Bacterial protein extraction was based on a previously published protocol [39]. Briefly, Falcon tubes (50 mL) with samples of *H. alvei* cultures were centrifuged at 3000× *g* for 20 min at 4 °C and supernatant was discarded. To wash out secreted compounds, bacterial pellets were resuspended with 0.1 M phosphate-buffered saline (PBS) and centrifuged at 3000× *g* for 20 min at 4 °C, the supernatant was discarded. Washes were repeated 3 times. To promote lysis, the pellets were frozen overnight at −20 °C. Bacterial cells were lysed in 2 mL R2D2 buffer (7 M urea, 2 M thiourea, 2% CHAPS, 0.05% Triphosphobutyl, 20 mM dithiothreitol, 0.5% C7BzO) with sonication for 6 min at 23% amplitude (3 s. ON/1 s. OFF). Addition of thiourea in the lysis buffer was aimed at promoting a better protein solubilization [40]. Soluble proteins were recovered by taking supernatant after centrifugation at 10,000× *g* for 30 min at 4 °C and quantified using Bradford assay. Protein extracts were diluted in saline (5 µg in 200 µL), a single dose per mouse, aliquoted and stored at −20 °C for further use.

### 4.3. Experiment #1 (Exp #1)

#### 4.3.1. Animals

The experimental procedures of the Exp# 1 were approved by the Animal Care and Use Committee at the Pavlov Institute of Physiology (Animal Welfare Assurance #A5952-01). Protocols were designed in accordance with the National Institutes of Health Guidelines for the Care and Use of Laboratory Animals. Subjects were adult mice of the inbred strain, C57BL/6J, derived from the parental stock obtained from the Jackson Laboratory (Bar Harbor, ME, USA). Experiments were performed with male mice 5–8 months of age, maintained at the vivarium of the Pavlov Institute. Animals were kept in standard polycarbonate cages on wood shaving bedding in a temperature- and humidity-controlled room (22–24 °C, 40–50%) with a 12/12 h light/dark cycle. During the acclimation period and throughout Exp #1 (Figure 1), mice were fed with a standard laboratory chow (PK-120, MEST Ltd., Moscow, Russia) containing 67% carbohydrates, 3% lipids, and 19% proteins, with an energy value of 2.99 kcal/g. Food and tap water were always available ad libitum.

#### 4.3.2. Probiotic Treatment and Experimental Design

Before starting the probiotic treatment, mice were isolated for 7 days in individual cages and had 2 training sessions for sweet preference test (see below) at 1-day intervals (Figure 1). Then, mice recevied a suspension of *H. alvei* bacteria (4.5 × 10^7^ CFU in 200 µL of saline/mouse) by intragastric gavage, or a total protein extract of *H. alvei* (5 µg/mouse in 200 µL of saline) daily at 4 pm for 17 days (0–16). Control group received 200 µL of saline. Taste preference for sucrose was assessed on days 11 and 12, as described below. The insulin tolerance test (ITT) was performed on day 7. Intraperitoneal and intragastric glucose tolerance tests (IP- and IG-GTT, respectively) were performed on days 9 and 15, respectively. On day 17, mouse tissues were collected under terminal sedation as described below in detail.

#### 4.3.3. Sweet Preference Test

The brief-access licking test (BALT) with sweet solutions was conducted in the middle of the light period (11 a.m.–3 p.m.) using procedures described by Glendinning et al. [41]. During the test, the mouse was placed in the chamber of the Davis MS-160 gustometer (DiLog Instruments, Tallahassee, FL, USA), in which a sipper tube containing test solution was presented 24 times for 5 s, with a 20-s interval. Animals first had two training sessions with water at 1-day intervals. On training day 1, they could drink water freely for 30 min from a single spout. On training day 2, the mouse had 24 trials with water, each lasting for 5 s. After each training/test day, animals had free access to water for 60 min in their home cages. To motivate licking during the training session, mice were deprived of water for 22–23 h before training. To stimulate drinking of sweet solutions, we limited mice to 1.5 mL of water for 22–23 h prior the test session. In the BALT, animals had access to sucrose solutions (Vecton Ltd., St. Petersburg, Russia) dissolved in deionized water at two ranges of concentration: 0.5–4% and 8–32%. The sequence of concentrations during the test was the same for all animals.

#### 4.3.4. Glucose and Insulin Tolerance Tests

Basal glucose assay as well as glucose and insulin tolerance tests were performed in the middle of the light period in non-fasted, awake mice in their home cages. The choice of using non-fasted animals was based on our objective to not interfere with the daily *H. alvei* administration. Moreover, mice typically do not consume food during the light phase, while the utility of performing the GTTs during light phase in non-fasted mice was validated in our previous study [42]. In the GTT, in accordance with specific recommendations of the National Institutes of Health Mouse Metabolic Phenotyping Center (McGuinness et al. 2009), a dose of 2.0 g/kg glucose (Sigma Aldrich, Burlington, MA, USA ) was administered in aqueous solution either by IG gavage or IP injection.

In the ITT, animals were injected with IP with 1.0 U/kg insulin (Actrapid^®^ HM, Novo Nordisk A/S, Bagsvaerd, Denmark). Blood was sampled via tail incision, and measurements of glucose concentration were made at 0, 10, 15, 30, 60, 90, and 120 min after injection of glucose or at 0, 15, 60, and 120 min after administration of insulin, using a handheld glycemic meter Contour PlusTM One (Ascensia Diabetes Care Holdings AG, Basel Switzerland).

#### 4.3.5. Body Composition and Insulin Assay

Mice were terminally sedated with a gas mixture of CO_2_ and O_2_ (50/50 volume %). Blood was withdrawn via aortic puncture into tubes containing EDTA and centrifuged for plasma separation, frozen and kept at −30 °C. Non-fasted insulin assay was performed using an ELISA kit CEA448Mu (Cloud-Clone Corp., Katy, TX, USA). Liver, pancreas, and fat were removed and weighed to the nearest 0.001 g. The anterior subcutaneous (interscapular), posterior subcutaneous (dorsolumbar, inguinal, and gluteal), visceral perirenal, visceral mesenteric, and retroperitoneal visceral epididymal (gonadal) bilateral fat depots were excised. Glycogen concentration in liver and muscle (*M. vastus medialis*) was measured spectrophotometrically [43]. Brains were removed and the hypothalamus was dissected and kept in IntactRNA (Evrogen, Moscow, Russia) formalin-based stabilizing reagent at −30 °C until further extraction.

#### 4.3.6. Real-Time Quantitative Reverse Transcription Polymerase Chain Reaction (qRT)-PCR

Total RNA was isolated from the hypothalamic tissue using a TRIzol analogue the ExtractRNA Reagent (Evrogen, Moscow, Russia) according to the manufacturer’s protocol. Samples containing 2 μg of RNA were subjected to reverse transcription to cDNA using the MMLV RT kit (Evrogen) and the oligo(dT) oligodeoxynucleotide primers. PCR amplification was performed using a mixture (reaction volume of 10 μL) containing 160 ng of the RT product, 20 pM of forward and reverse primers, and the qPCRmix-HS SYBR kit (Evrogen). The amplified signals were continuously detected using the CFX96 RealTimeSystem (Bio-Rad Laboratories, Inc., Hercules, CA, USA). The following PCR amplification protocol was used: (i) initial denaturation at 95 °C for 5 min; (ii) a 3-segment amplification and quantification program consisting of 45 cycles of 95 °C for 5 s, 57–61 °C for 5 s, and 72 °C for 10 s (data were collected during this step); and (iii) the ABI Melt Curve program to check for one peak and no primer dimer formation in each reaction containing the template. A list of primers is present in Table 1. 

Due to the expressions of house-keeping gene-encoding the TATA-binding protein was used as an endogenous standard for estimation of neuropeptide expression. The results were analyzed using the delta-delta Ct method (https://toptipbio.com/delta-delta-ct-pcr, accessed on 16 January 2023) and data are shown as fold levels relative to the expression in the control group of mice.

### 4.4. Experiment # 2 (Exp#2)

#### 4.4.1. Animals

The experimental procedures of Exp #2 were approved by the Normandy Regional Ethics Committee (n. 6701). Animal manipulations were performed according to the European Community Council Directive of 24 November 1986 (86:609: EEC). C57Bl/6J 8 weeks old male mice from Janvier Laboratory (Le Genest-Saint-Isle, France) were used in this study. Mice were housed with free access to a standard laboratory diet (UAR, Villemoisson-sur-Orge, France) and water. They were kept in a ventilated room at a temperature of 22  ±  1 °C under a 12 h light/12 h dark cycle (light on between 07:00 and 19:00). All the experiments were carried out between 9.00 h and 18.00 h. 

#### 4.4.2. ClpB Administration and Glucose Measurements in Mice

To determine the effect of ClpB on basal plasma glucose, mice were fasted for 6 h with free access to water before the ClpB injection. For the IP-GTTs and IG-GTTs, mice were fasted for 16 h with free access to water. Recombinant *Escherichia coli (E. coli)* ClpB protein was custom made by Delphi Genetics (Gosselies, Belgium) for Targedys SA (Longjumeaux, France) who kindly provided it for this study. *E. coli* ClpB is highly homologous to *H. alvei* ClpB, having the same α-MSH-like epitope [10]. ClpB protein was dissolved in a sterile PBS buffer. Each mouse received 10.0 μg ClpB via IP injection in a total volume of 100 μL PBS 30 min before 2.0 g/kg glucose administration via IP injection or IG gavage during the IP-GTT and IG-GTT, respectively, *n* = 8. Control group (*n* = 8) received the same volume of PBS alone. The same mice were analyzed for ClpB effects on basal glucose as well as in IP- and IG-GTT with one-week interval between the tests. Blood sampling was obtained via tail vein incisions every 15 min during 2 h. Blood glucose concentrations were measured using an Accu-Chek Performa glucometer (Roche Diagnostic, Saint-Égrève, France).

#### 4.4.3. Statistical Analysis

The data from all experiments are presented as mean ± SEM with *p*-values < 0.05 considered as significant. Statistical analysis in Exp#1 was performed using Statistica 7.0 software (StatSoft, Tulsa, OK, USA) and graphs were plotted with Excel (Microsoft Corp.). In Exp#2 data were analyzed and graphs plotted using GraphPad Prism 9.0 (GraphPad Software, San Diego, CA, USA). Baseline glucose, plasma insulin and body composition parameters were compared using one-way ANOVA and Tukey’s HSD post hoc test. Data of GTT, ITT, and taste responses were compared using one- or two-way ANOVA. For two-way ANOVA, concentration (for taste tests) and time (for GTT or ITT) were considered as within-subject factors, and effect of gavage as a between-subject factor. Post hoc paired comparisons were made using the Fisher’s least significant difference (LSD) test. The area under the curve (AUC) was calculated using the trapezoidal rule. Differences between AUCs for glucose or water consumption in training sessions of the BALT were determined with one-way ANOVA. rtPCR data were compared using Mann–Whitney’s U-test. Other comparisons were made with unpaired two-tailed Student’s t-test.

## 5. Conclusions

In conclusion, we should first recapitulate some of the main limitations of this study, such as that we did not explore whether the obtained effect on glucose metabolism induced by *H. alvei* supplementation was strain-specific or whether it was due to the presence of ClpB protein. The molecular mechanisms underlying this effect also remain undetermined. Nevertheless, our study is of practical relevance because it reveals that the ability of the total protein extract of *H. alvei* 4597 strain to increase glucose tolerance is apparently independent from the plasma-glucose-lowering effect of the same strain of live bacteria. Indeed, the results of our study support the idea that, while the protein extract acts preferentially at the glucose postabsorptive level, live bacteria may induce an incretin effect. These data further suggest that a combination of both *H. alvei* bacteria and their protein extract should be tested in animal models of diabetes and obesity. Finally, we found that the enterobacterial ClpB protein, as well as its known satietogenic action, has the additional beneficial effect of increasing glucose tolerance, which can be exploited for the prevention of metabolic diseases.

## Figures and Tables

**Figure 1 ijms-24-10590-f001:**
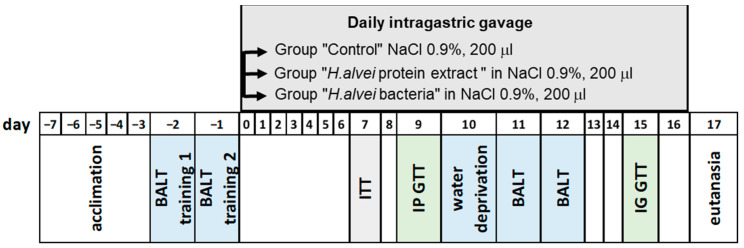
Schedule of Experiment #1.

**Figure 2 ijms-24-10590-f002:**
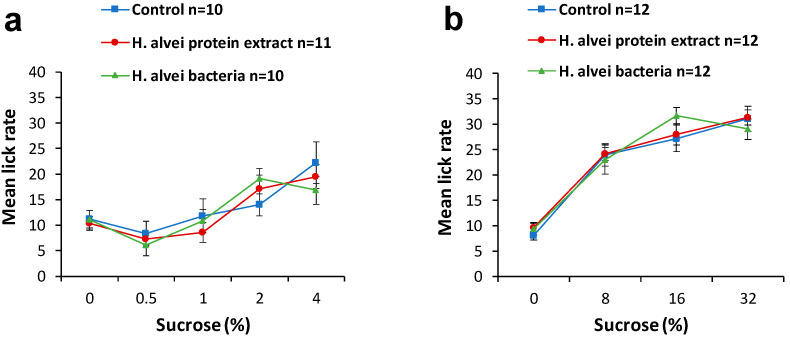
Licking responses to solutions with low and high ranges of sucrose concentrations, <4.0% (**a**) and <32.0% (**b**), respectively, in the brief-access licking test of C57BL/6J lean mice pretreated for 11–12 days via gavage with suspension of *H. alvei* (daily dose of 4.5 × 10^7^ CFU/mL) or protein extract of *H. alvei* (daily dose 5 μg).

**Figure 3 ijms-24-10590-f003:**
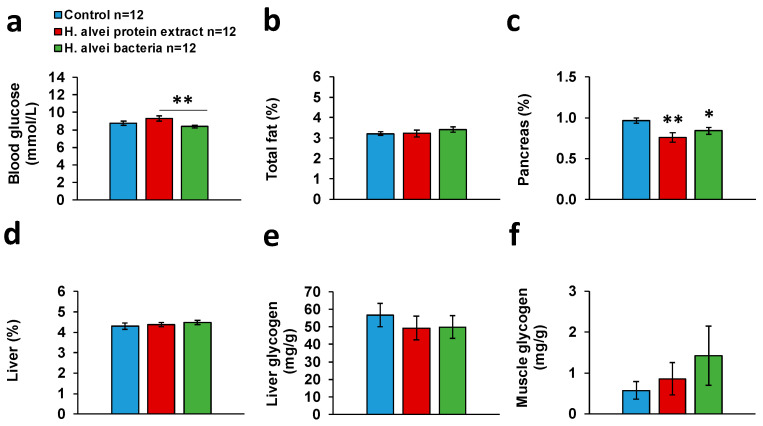
Effect of a 17-day gavage of C57BL/6J mice with suspension of bacteria or protein extract of *H. alvei* on basal blood glucose concentration (**a**), body fat content (**b**), pancreas mass (**c**), and liver mass (**d**) relative to 100% of body weight, glycogen concentration in the liver (**e**) and muscle (**f**). ** *p* < 0.01, * *p* < 0.05, Tukey HSD post hoc test vs. control, unless specified.

**Figure 4 ijms-24-10590-f004:**
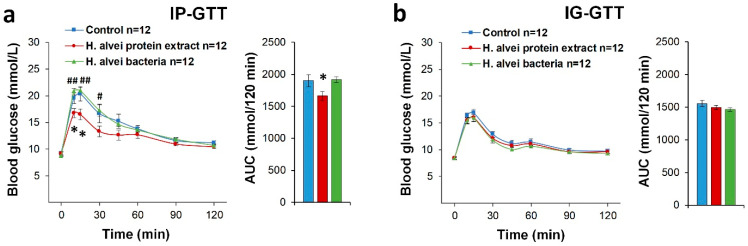
Effect of chronic provision of a suspension of *H. alvei* bacteria (4.5 × 10^7^ CFU/mL daily) or their total protein extract (5 µg daily) on glucose tolerance in lean, non-fasted C57BL/6J mice as compared to controls receiving saline. Dynamics of plasma glucose levels and AUC after intraperitoneal or intragastric administration of 2.0 g/kg glucose in the IP-GTT (**a**) and IG-GTT (**b**), respectively. (**a**) Post hoc comparisons with Fisher LSD test; * *p* < 0.05, Protein vs. Control, ^#^
*p* < 0.05 and ^##^ *p* < 0.01 Protein vs. Bacteria. Color code for AUC is the same as for the curves.

**Figure 5 ijms-24-10590-f005:**
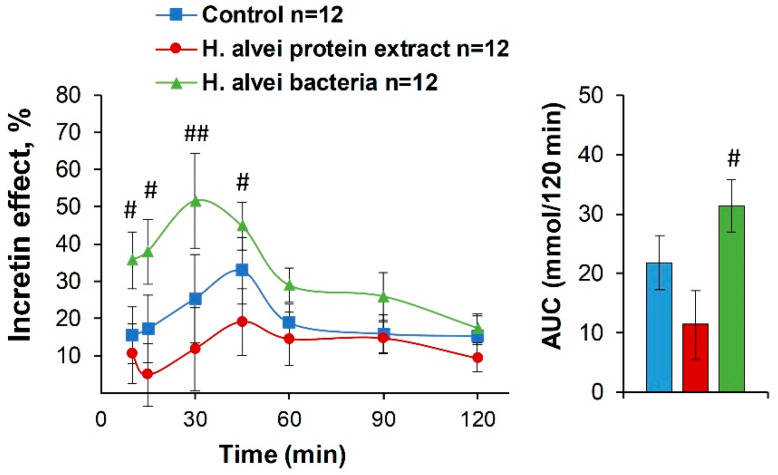
‘Incretin effect’ evaluated as a % of differences of plasma glucose between IP-GTT and IG-GTT vs. 100% IG-GTT glucose levels for each time-point. Post-hoc comparisons for AUC with Fisher’s LSD test, bacteria vs. protein groups, ^##^
*p* < 0.01, *p* < 0.05. ^#^
*p* < 0.05. Color code for AUC is the same as for the curves.

**Figure 6 ijms-24-10590-f006:**
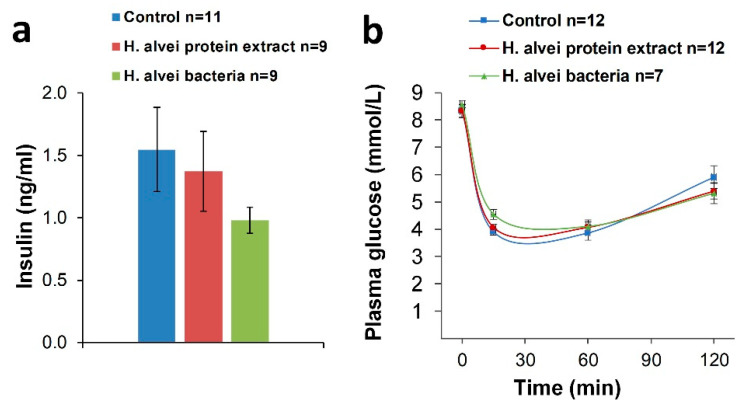
Effect of intragastric administration of *H. alvei* bacterial suspension and *H. alvei* protein extract on plasma insulin level and insulin tolerance. (**a**) Basal plasma concentrations of insulin after 17 days of probiotic treatment; (**b**) IP insulin tolerance test (ITT); plasma glucose levels after IP administration of insulin (1U/kg).

**Figure 7 ijms-24-10590-f007:**
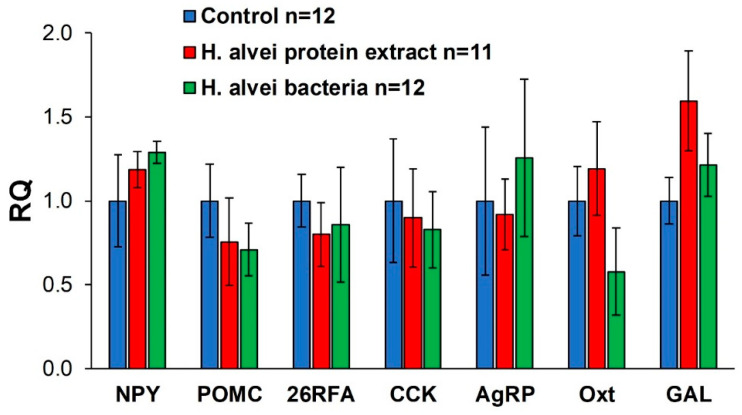
Expression of hypothalamic regulatory peptide mRNA of mice receiving bacterial suspension or protein extract of *H. alvei* for 17 days. Control mice were pretreated with saline. The data are presented as a relative quantity (RQ = log(2^−ddCt^)) to the control group. NPY—neuropeptide Y, POMC—proopiomelanocortin, GAL—galanin, 26RFa—neuropeptide 26RFA, CCK—cholecystokinin, AgRP—agouti-related protein, Oxt—oxytocin.

**Figure 8 ijms-24-10590-f008:**
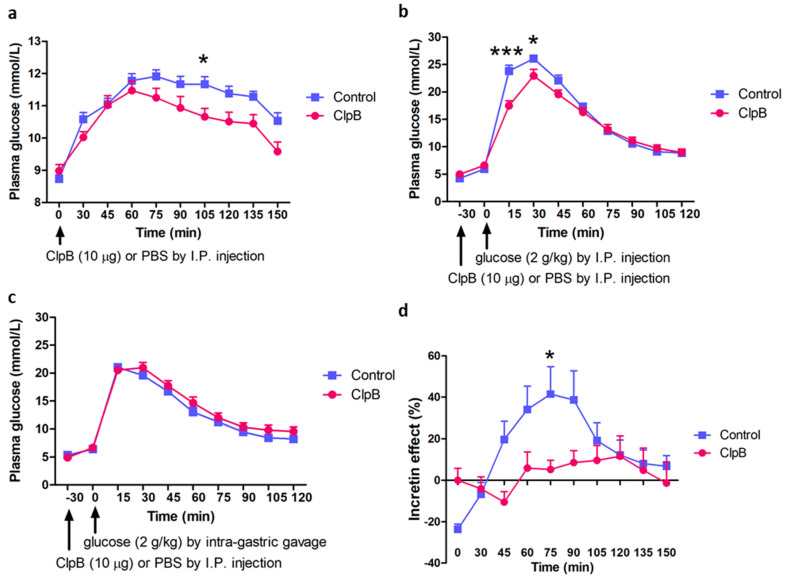
Effect of acute IP administration of 10.0 μg of the recombinant ClpB protein on plasma glucose in mice in basal conditions (**a**) or during the IP-GTT (**b**) and IG-GTT (**c**). ‘Incretin effect’ is shown as the % of differences of plasma glucose between IP and IG vs. 100% IG glucose levels (**d**). 2-way ANOVA, Bonferroni, post-tests * *p* < 0.05, *** *p* < 0.001, *n* = 8 for both groups.

**Table 1 ijms-24-10590-t001:** PCR primer sequences.

Gene Target	Primer Sequences (Forward and Reverse)	PCR Product Length in Base Pairs (b.p.) and T_ann_
Agouti-related protein (AgRP)	5′-CCCAGAGTTCCCAGGTCTAAGTCT-3′ 5′-CACCTCCGCCAAAGCTTCT-3′	100 b.p. 61 °C
Beta-actin	5′-TCCACACCCGCCACCAGTTC-3′ 5′-GGAGCATCGTCGCCCGC-3′	103 b.p. 59 °C
Cholecystokinin (CCK)	5′-GCTGATTTCCCCATCCAAA-3′ 5′-GCTTCTGCAGGGACTACCG-3′	105 b.p. 58 °C
Galanin (Gal)	5′-CACAGATCATTTAGCGACAAGCAT-3′ 5′-GACAATGTTGCTCTCAGGCAG-3′	114 b.p. 59 °C
Neuropeptide 26RFA	5′-GAAGGGGACCCACAGACATC-3′ 5′-GTCTTGCCTCCCTAGACGGAA-3′	176 b.p. 60.5 °C
Neuropeptide Y (NPY)	5′-CCGCTCTGCGACACTACAT-3′ 5′-TGTCTCAGGGCTGGATCTCT-3′	68 b.p. 60 °C
Oxytocin (Oxt)	5′-GACCTGGATATGCGCAAGTGT-3′ 5′-GAAGCAGCCCAGCTCGTC-3′	96 b.p. 60 °C
Pro-opiomelanocortin (POMC)	5′-CAGTGCCAGGACCTCACC-3′ 5′-CAGCGAGAGGTCGAGTTTG-3′	72 b.p. 59 °C
TATA-binding protein	5′-CTGCTGTTGGTGATTGTTGGT-3′ 5′-AGGCGGAATGTATCTGGCAC-3′	199 b.p. 59 °C

## Data Availability

Data are available upon request.

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
