# Peer review of "Protein Extract of a Probiotic Strain of Hafnia alvei and Bacterial ClpB Protein Improve Glucose Tolerance in Mice"

_ijms, 2023, doi:10.3390/ijms241310590_

Round 1

Reviewer 1 Report

This is an interesting paper on the improvement of glucose tolerance in mice by the protein extract of Hafnia alvei and bacterial ClpB protein. However I would like to draw your attention to the following:

In the abstract you state Mice treated with the H. alvei protein extract showed an improved glucose tolerance in IP-GTT but not in IG-GTT. And then

H. alvei protein extract introduced orally and the ClpB protein by IP injection, both improve glucose tolerance acting probably at the postabsorptive level.

 These two sentences are contradictory. Please amend accordingly.

Methods Line 103.   You state that Hafnia alvei 4597 was cultured for 24h and you determine CFU/mL at 6h only. Samples were diluted in 0.9% NaCl to 4.5x107 in 200 μl, a single dose per mouse, aliquoted and stored at -20°C for further use. The time point of samples collection is not clear. Moreover, sine no growth curve is provided, it is not clarified during which growth phase you collected your samples. Since the growth phase greatly influences protein synthesis, hence protein concentration, clarification is needed.  

 Methods Line 106. Is the described protocol for protein extraction a published protocol? Please provide relevant references.   Did 7M urea affect your proteins? Does the term diluted proteins refer to soluble proteins?

 Line 132 Figure 1.  In this figure it appears that all individuals received both gavage, IP GTT and IG GTT. However you state that mice received daily at 4 p.m. for 17 days (0–16) by intragastric gavage of a suspension of H. alvei bacteria (4.5×107 CFU in 200 μL of saline/mouse), or an extract of H. alvei total protein (5 μg/mouse in 200 μL of saline).

You might need to add more lines to the Table of Figure 1, to discern among experimental groups, which are not clearly stated.

 Line 204. Please state in the introduction the relevance of the particular gene targets in the present study.

 Line 384 Did you actually perform IP-GTT and IG-GTT one week later on the same individuals?

 Line 389 These results suggest that some bacterial proteins derived from the total protein extract H. alvei may stimulate insulin secretion independently from the intestinal incretins. Because such an effect was not present in mice receiving the H. alvei bacteria, it suggests that alive bacteria do not release sufficient amount of proteins in the upper gut.

This appears slightly presumptuous. You administered proteins extracted from bacteria cells, these proteins have not been identified and may well include intrinsic cell proteins as well as proteins intended for secretion. Hence your conclusion needs to be restated.

Author Response

This is an interesting paper on the improvement of glucose tolerance in mice by the protein extract of Hafnia alvei and bacterial ClpB protein. However, I would like to draw your attention to the following: In the abstract you state Mice treated with the H. alvei protein extract showed an improved glucose tolerance in IP-GTT but not in IG-GTT. And then H. alvei protein extract introduced orally and the ClpB protein by IP injection, both improve glucose tolerance acting probably at the postabsorptive level.  These two sentences are contradictory. Please amend accordingly.

Response: We thank the Reviewer for this and other comments which were most useful. We apologize for this sentence in the conclusion which was misleading, in fact we have meant that ClpB and H.alvei protein extract are acting probably at the post-absorptive level of glucose. To clarify this issue, the corresponding sentences in the abstract and discussion have been modified.

Methods Line 103.You state that Hafnia alvei 4597 was cultured for 24h and you determine CFU/mL at 6h only. Samples were diluted in 0.9% NaCl to 4.5x107 in 200 μl, a single dose per mouse, aliquoted and stored at -20°C for further use. The time point of samples collection is not clear. Moreover, since no growth curve is provided, it is not clarified during which growth phase you collected your samples. Since the growth phase greatly influences protein synthesis, hence protein concentration, clarification is needed.

Response:

The Reviewer is right that the protein production depends on the bacterial growth phase. This phenomenon was, indeed, considered during the experimental design and the choice of 6h for the sample preparation was based on the beginning of the stationary phase in the culture conditions used. The bacterial growth dynamics were verified. To answer to this comment, this point has been clarified in the “methods” section.

 Methods Line 106. Is the described protocol for protein extraction a published protocol? Please provide relevant references. Did 7M urea affect your proteins? Does the term diluted proteins refer to soluble proteins?

Response:

The protocol of bacterial protein extraction was based on the previously published protocol (Breton et al 2016) modified to improve its yield previously validated in our laboratory for H.alvei. The addition of 2M thiourea in the lysis buffer, helped to improve protein solubilization (Peach et al 2012). And, yes ‘diluted’ meant ‘soluble’, thank you for noticing it. The methods are now clarified accordingly and relevant references are cited.

 Line 132 Figure 1.  In this figure it appears that all individuals received both gavage, IP GTT and IG GTT. However, you state that mice received daily at 4 p.m. for 17 days (0–16) by intragastric gavage of a suspension of H. alvei bacteria (4.5×107 CFU in 200 μL of saline/mouse), or an extract of H. alvei total protein (5 μg/mouse in 200 μL of saline). You might need to add more lines to the Table of Figure 1, to discern among experimental groups, which are not clearly stated.

Response:

The figure 1 has been modified to show different groups.

 Line 204. Please state in the introduction the relevance of the particular gene targets in the present study.

Response: a sentence justifying the choice of regulatory peptide gene targets with corresponding references has been included in the introduction.

 Line 384 Did you actually perform IP-GTT and IG-GTT one week later on the same individuals?

Response: Yes, these were the same mice as specified in the Methods section. To make it more clear, this information has now been included also in the corresponding Results section.

 Line 389 These results suggest that some bacterial proteins derived from the total protein extract H. alvei may stimulate insulin secretion independently from the intestinal incretins. Because such an effect was not present in mice receiving the H. alvei bacteria, it suggests that alive bacteria do not release sufficient amount of proteins in the upper gut. This appears slightly presumptuous. You administered proteins extracted from bacteria cells, these proteins have not been identified and may well include intrinsic cell proteins as well as proteins intended for secretion. Hence your conclusion needs to be restated.

Response: We agree with this comment and have modified the corresponding sentence accordingly.

Reviewer 2 Report

In this paper, the Authors aimed to demonstrate if H. alvei protein extract, orally administrered, or the ClpB protein, injected by IP and IG-GTT, is able to i) improve glucose tolerance in C57BL/6J mice and ii) modify sweet taste preferences.

H. alvei protein extract showed an increased glucose tolerance only in IP-GTT, a reduction of pancreatic weight, but no significant basal plasma insulin and sweet taste preference changes. 

The work is well done and the conclusions are already  exhaustive.

A few suggestions for Authors:

- Improve the quality of the English language/grammar.

- Check for bibliographic references. All the cyted journal names should be abbreviated (some of them are not).

The quality of the English language/grammar shoud be improved.

Author Response

In this paper, the Authors aimed to demonstrate if H. alvei protein extract, orally administered, or the ClpB protein, injected by IP and IG-GTT, is able to i) improve glucose tolerance in C57BL/6J mice and ii) modify sweet taste preferences.

  1. alvei protein extract showed an increased glucose tolerance only in IP-GTT, a reduction of pancreatic weight, but no significant basal plasma insulin and sweet taste preference changes.

The work is well done and the conclusions are already exhaustive.

A few suggestions for Authors:

- Improve the quality of the English language/grammar.

Response: We thank the Reviewer for the appreciation of our study and nice comments. The manuscript has been carefully checked for the language/grammar.

- Check for bibliographic references. All the cited journal names should be abbreviated (some of them are not).

Response: The reference list was checked for the consistency of using standard abbreviation of the journals’ names.

Round 2

Reviewer 1 Report

It was a pleasure to see that all points raised have been addressed